# Time-Lapse Flow Cytometry: A Robust Tool to Assess Physiological Parameters Related to the Fertilizing Capability of Human Sperm

**DOI:** 10.3390/ijms22010093

**Published:** 2020-12-24

**Authors:** Arturo Matamoros-Volante, Valeria Castillo-Viveros, Paulina Torres-Rodríguez, Marcela B. Treviño, Claudia L. Treviño

**Affiliations:** 1Departamento de Genética del Desarrollo y Fisiología Molecular, Instituto de Biotecnología, Universidad Nacional Autónoma de México, Cuernavaca Morelos 62210, Mexico; arturo.matamoros@mail.ibt.unam.mx (A.M.-V.); linda.castillo@mail.ibt.unam.mx (V.C.-V.); torresp@ibt.unam.mx (P.T.-R.); 2Science Department, School of Pure and Applied Sciences, Florida SouthWestern State College, Fort Myers, FL 33919, USA; marcela.trevino@fsw.edu

**Keywords:** time-lapse flow cytometry, membrane potential, intracellular calcium, intracellular pH, sperm capacitation, sperm fertilizing capability

## Abstract

Plasma membrane (PM) hyperpolarization, increased intracellular pH (pH_i_), and changes in intracellular calcium concentration ([Ca^2+^]_i_) are physiological events that occur during human sperm capacitation. These parameters are potential predictors of successful outcomes for men undergoing artificial reproduction techniques (ARTs), but methods currently available for their determination pose various technical challenges and limitations. Here, we developed a novel strategy employing time-lapse flow cytometry (TLFC) to determine capacitation-related membrane potential (*E*_m_) and pH_i_ changes, and progesterone-induced [Ca^2+^]_i_ increases. Our results show that TLFC is a robust method to measure absolute *E*_m_ and pH_i_ values and to qualitatively evaluate [Ca^2+^]_i_ changes. To support the usefulness of our methodology, we used sperm from two types of normozoospermic donors: *known paternity* (subjects with self-reported paternity) and *no-known paternity* (subjects without self-reported paternity and no known fertility problems). We found relevant differences between them. The incidences of membrane hyperpolarization, pH_i_ alkalinization, and increased [Ca^2+^]_i_ were consistently high among *known paternity* samples (100%, 100%, and 86%, respectively), while they varied widely among *no-known paternity* samples (44%, 17%, and 45%, respectively). Our results indicate that TLFC is a powerful tool to analyze key physiological parameters of human sperm, which pending clinical validation, could potentially be employed as fertility predictors.

## 1. Introduction

Sexual reproduction involves the successful fusion of the female (egg or oocyte) and male (sperm) gametes, a process called fertilization. Human fertility-related issues are a growing public health problem worldwide [1]. The causes of human infertility are multifactorial and the prevalence among females and males occurs in the same proportion [2,3]. Regarding male factors, sperm dysfunctions are considered to be the most frequent etiology of fertility issues [4]. The most common approach to detect such dysfunctions on sperm is through the evaluation of macro- and microscopic seminal parameters (i.e., semen volume and pH, sperm motility, vitality, and morphology), which are collectively known as the seminogram [5]. The World Health Organization (WHO) updated the reference values for the seminogram analysis in 2010 [6]. Men who fulfill these reference values are considered normozoospermic and are therefore presumed to be fertile. However, the usefulness of these parameters as predictors of reproductive outcomes has been on debate for seven decades, given that the reference values observed in fertile men usually overlap with those obtained in men with fertility issues [6,7,8]. Thus, the sole evaluation of seminogram parameters is not sufficient to unequivocally establish whether a male individual is fertile or infertile [9,10].

As a result, various efforts have been made to correlate the fertilizing capability of sperm with some of the physiological events that take place during sperm capacitation, a multistep phenomenon involving changes in sperm form and function [11], which is essential for fertilization, and which encompasses various molecular processes that sperm cells undergo during their transit inside the female reproductive tract [12,13,14]. Among these processes we can distinguish (a) changes in the plasma membrane’s ion permeability, which result in its hyperpolarization ([15,16,17], reviewed in [18]); (b) an increase in intracellular pH (pH_i_) [19,20,21]; and (c) a rise in intracellular calcium concentration ([Ca^2+^]_i_) upon exposure to progesterone (Pg, a female hormone secreted by cumulus *oophorus* cells that surround the oocyte) [22,23,24]. These parameters have been independently analyzed in sperm samples from men with fertility issues, and results show that after capacitation induction, their sperm cells do not undergo pH_i_ alkalinization, and they also fail to hyperpolarize (in some cases they depolarize) their plasma membrane (PM) [17,25,26,27,28]. Regarding [Ca^2+^]_i_ changes, some reports suggest that sperm cells from men with fertility issues present a decreased response to Pg [29,30]. Collectively, these shreds of evidence support the notion that the evaluation of sperm capacitation-related parameters may potentially serve as a tool to predict outcomes for different artificial reproduction techniques (ARTs), which may in turn help guide ART selection for men seeking to overcome fertility issues.

The methods currently employed to analyze the aforementioned capacitation-related parameters include single cell-based approaches, such as time-lapse fluorescence video-microscopy [29] and the electrophysiological patch-clamp technique [25,31,32]. These procedures present the disadvantage of only permitting the analysis of a limited number of cells, which can mask heterogeneous responses and therefore yield inaccurate results. Other methods include population-based approaches, such as spectrofluorometry [17,26] and flow cytometry [27,28], which analyze the behavior of an entire sperm cell sample population. Spectrofluorometry poses the disadvantage of only permitting the recording of average responses, and as a result, some non-homogeneous responses may also be masked. In contrast, flow cytometry evaluates single-cell responses for a large number of cells, thus enabling identification, if present, of cell subpopulations with heterogeneous responses. Furthermore, flow cytometry enables the simultaneous use of several markers to evaluate different cellular parameters. However, conventional flow cytometers are not fitted for the continuous addition of test compounds to the cell suspension since the pressurized system for sample aspiration requires the creation of a vacuum. When using these instruments, it is therefore necessary to pause aspiration in order to proceed with reagent addition. Due to the time delay between each addition, such stopped-flow cytometry methods are not useful for the performance of continuous measurements, which are required for the observation of fast and/or transient responses. The BD Accuri C6 Plus flow cytometers are suitable instruments to overcome these limitations because they operate with a peristaltic pump, eliminating the need for vacuum creation. Continuous compound additions are thus feasible through manual pipetting, enabling the possibility of conducting time-lapse fluorescence recordings for a large cell population, with single-cell resolution [33,34].

Taking advantage of this instrument’s unique feature, we developed a novel method, which we have named time-lapse flow cytometry (TLFC). Our main findings suggest that TLFC is suitable for measurements of both membrane potential (*E*_m_) and pH_i_ absolute values, as well as for the qualitative evaluation of [Ca^2+^]_i_ changes. We compared all three cellular responses in sperm cells incubated under conditions that either support or do not support in vitro capacitation. We also found promising preliminary results when comparing the incidence of the three capacitation parameters in sperm samples from normozoospermic donors with self-reported paternity (*known paternity*) versus donors with no self-reported paternity and no known fertility issues (*no-known paternity*). Altogether, our results lead us to propose TLFC as a robust method for the analysis of physiological parameters in human sperm cells, with the potential for use in male fertility evaluations and ART selection.

## 2. Results

### 2.1. Measurement of E_m_ Absolute Values in Human Sperm Using TLFC Recordings

PM hyperpolarization occurs during sperm capacitation [15,35,36], and this physiological event has been suggested as a predictor of success output during ART implementation [26,27]. For this reason, we established a strategy to evaluate the *E*_m_ of human sperm employing TLFC using 3,3′dipropylthiadicarbocyanine iodide (DiSC_3_(5), abbreviated to Disc), a cationic carbocyanine *E*_m_-sensitive probe. This fluorescent dye partitions into the sperm PM according to its *E*_m_—when the PM becomes hyperpolarized, Disc will accumulate in the PM due to its cationic nature, while a depolarization favors the dye’s efflux from the cell, resulting in decreased fluorescence [37]. Taking advantage of the multiparameter analysis that flow cytometry permits, we loaded sperm samples with 4-(6-acetoxymethoxy-2,7-dichloro-3-oxo-9-xanthenyl)-4′-methyl-2,2′(ethylenedioxy) dianiline-N,N,N′,N′-tetra acetic acid tetrakis (acetoxymethyl) ester (Fluo3-AM, abbreviated to Fluo3) prior to the experiment. This Ca^2+^-sensitive probe exhibits an increase in fluorescence upon binding Ca^2+^, indirectly allowing the exclusion of non-viable cells from the analysis (see Section 4.5 and Appendix A for details). After co-staining each sperm sample with Disc, we performed fluorescence recordings in an BD Accuri C6 Plus flow cytometer retrofitted with our custom-built system for the continuous addition of reagents (see Section 4.4 and Figure 1).

First, resting-state Disc fluorescence was recorded for 120 s. This was immediately followed by the addition of the K^+^ ionophore valinomycin (1 μM), which evoked pharmacological clamping of *E*_m_ by transforming the sperm PM into a “K^+^ electrode”. This resulted in a net movement of K^+^ ions out of the cell until the K^+^ reversal potential was reached, causing membrane hyperpolarization, which in turn brought about the accumulation of Disc in the lipid bilayer of the PM and the concomitant rise in PM fluorescence. After reaching a fluorescence plateau, sequential additions of KCl were made to elicit K^+^ influx and therefore trigger membrane depolarization (KCl was added in the amounts required to reach the *E*_m_ target values of −80, −63, −52, −40, and −28 mV) (Figure 2a). Then, the median fluorescence value for the responding sperm subpopulation (indicated in gray boxes in Figure 2a) when the plateau was reached was plotted against the theoretical *E*_m_ value obtained by applying the Nernst’s equation (Figure 2b). A linear model was fitted to these data, and fluorescence values were interpolated to calculate the corresponding resting *E*_m_ values (Figure 2b,c). In this experiment, which was used as a representative example, the calculated resting *E*_m_ value for the sperm subpopulation was −65 mV (Figure 2c, gray histogram). This resting *E*_m_ value is consistent with previously reported data [26,27], strongly suggesting that our TLFC method is suitable for measuring absolute *E*_m_ values in the human sperm model. As a proof of concept, we employed this method to evaluate the *E*_m_ of sperm samples incubated in either non-capacitating (NC) or capacitating (CAP) conditions. We analyzed samples from two distinct classes of normozoospermic donors: (i) *known paternity* subjects, who had self-reported paternity (*n* = 6), and (ii) *no-known paternity* subjects, who had no self-reported paternity and no known fertility issues (*n* = 9). In concordance with previous reports where *E*_m_ was measured using flow cytometry [26,27], we found a high level of heterogeneity in resting *E*_m_ values across samples, especially among the *no-known paternity* group. Actually, in samples from this donor class, we observed only a slight and non-statistically significant difference in resting *E*_m_ between NC and CAP conditions (−67 ± 4 and −70 ± 3 mV, respectively; *p* = 0.25). Interestingly, we detected a capacitation-induced hyperpolarization in all *known paternity* samples (−60 ± 3 versus −71 ± 3 mV for NC and CAP conditions, respectively; *p* = 0.01) (Figure 2d). To further analyze these data, we calculated the average *E*_m_ changes upon capacitation (Δ*E*_m_ = *E*_mCAP_ − *E*_mNC_) (Figure 2e). As previously reported [26], a Δ*E*_m_ = ± 5 mV was established as the threshold value to consider a sperm sample as either undergoing PM depolarization (Δ*E*_m_ > 5 mV) or hyperpolarization (Δ*E*_m_ < 5 mV). Using these criteria, we found that 44% of the *no-known paternity* samples were either hyperpolarized or unchanged upon capacitation, while only 1 out of 9 samples (12%) underwent depolarization. In contrast, 100% of the *known paternity* samples displayed the expected capacitation-associated hyperpolarization (Figure 2f).

### 2.2. Simultaneous Measurement of E_m_ and Evaluation of [Ca^2+^]_i_ Changes in Human Sperm Using TLFC Recordings

As mentioned before, our TLFC measurements of *E*_m_ were performed by employing cells stained with two different dyes—Fluo3 and Disc. Such co-staining capability offers an opportunity for the simultaneous evaluation of more than one physiological parameter, up to four, provided that the required fluorescent dyes are compatible. In this case, the combined use of Fluo3 and Disc allowed us to simultaneously analyze changes in [Ca^2+^]_i_ during the protocol for *E*_m_ measurements. Interestingly, in most of the sperm samples tested, Fluo3 fluorescence rapidly dropped after PM hyperpolarization with valinomycin, reaching a minimum after 60 s, followed by a recovery phase (Figure 3a,b). Then, in all sperm samples analyzed, depolarization pulses triggered by sequential KCl additions evoked a transient [Ca^2+^]_i_ increase. For comparison purposes, we normalized fluorescence values by setting the basal levels to zero, which was accomplished by subtracting the average of the basal level (F_0_) from each value fluorescence (F) and dividing by F_0_ (Figure 3c).

### 2.3. Qualitative Analysis of [Ca^2+^]_i_ Changes in Human Sperm Using TLFC Recordings

The Pg-mediated opening of the sperm-specific cationic channel CatSper generates an [Ca^2+^]_i_ increase [21,22], which is involved in cellular processes essential for fertilization, such as motility changes and stimulation of the acrosomal reaction (AR) [38,39]. The CatSper channel is a multimeric protein complex composed of nine different subunits: CatSper1–4 are pore-forming subunits and the other five are distinct auxiliary subunits (β, γ, δ, ε, and ζ) [40]. Interestingly, men harboring mutations or deletions on CatSper1- or CatSper2-encoding genes are infertile [41,42], and sperm samples from men undergoing ARTs present diminished responses to Pg compared to control samples from normozoospermic men [25,29]. For these reasons, we decided to test the suitability of TLFC for detecting the Pg-induced raise in [Ca^2+^]_i_.

Sperm samples were loaded with Fluo3 to follow [Ca^2+^]_i_ variations. Basal Fluo3 fluorescence values were recorded during the first 120 s. Immediately after this, Pg (4 μM) was added to the sperm suspension, triggering a fluctuation in [Ca^2+^]_i_ as previously reported [24,43,44]. To obtain the maximum and minimum fluorescence values, we added the Ca^2+^ ionophore ionomycin (10 μM) and manganese chloride (Mn^2+^, 5 mM), respectively, at the end of each recording (Figure 4a). As in the case of *E*_m_ measurements, we also evaluated the effect of Pg in sperm cells incubated under NC and CAP conditions. Figure 4b shows the normalized fluorescence values for comparison purposes. For each of the sperm samples from *known* (*n* = 7) and *no-known* (*n* = 11) *paternity* donors, we compared the normalized fluorescence peaks (F_Peak_) reached after Pg addition under NC and CAP conditions (Figure 4c). In samples from *known paternity* donors, we observed a greater and statistically significant response to Pg under CAP conditions compared to NC conditions (F_PeakNC_ = 0.88 ± 0.12 vs. F_PeakCAP_ = 1.90 ± 0.27, *p* = 0.0179). In contrast, we observed a wide range of responses among the samples from *no-known paternity* donors, and no statistically significant difference between NC and CAP conditions (F_PeakNC_ = 1.64 ± 0.34 vs. F_PeakCAP_ = 1.96 ± 0.36, *p* = 0.3646). To analyze the magnitude of the difference in F_Peak_ between NC and CAP conditions, we calculated ΔF_Peak_ = F_PeakCAP_ − F_PeakNC_ for each sample (Figure 4d). Considering that the standard error of the NC mean value for all 18 sperm samples was ±0.23 (data not shown), we arbitrarily established ΔF_Peak_ = ±0.25 as a threshold value to classify each individual sample as having either an increased, a decreased, or an unchanged Pg-induced change in [Ca^2+^]_i_ upon capacitation. This revealed that 86% of samples from *known paternity* donors exhibited an increased Pg response under capacitation conditions, and only one out of seven (14%) displayed no change. In contrast, 45% of samples from *no-known paternity* donors presented an increased response to Pg, 36% had no change, and 19% showed a decreased Pg response upon capacitation (Figure 4e).

### 2.4. Measurement of pH_i_ Absolute Values in Human Sperm Using TLFC

Intracellular alkalinization has been considered as a hallmark of the sperm capacitation process [45,46]. This physiological event is currently receiving attention from a clinical standpoint since a recent study reported that sperm samples from men undergoing ART did not display alkalinization [28]. In light of this, we decided to develop a protocol to determine absolute pH_i_ values in human sperm cells using TLFC. We employed the pH-sensitive dye 2′,7′-bis-(2-carboxyethyl)-5-(and-6)-carboxyfluorescein acetoxymethyl ester (BCECF-AM, abbreviated to BCECF), which exhibits increased fluorescence when the cytoplasm becomes alkalinized, and decreased fluorescence upon acidification.

The chemical nature of BCECF makes it suitable as a viability marker as well, similar to Fluo3 (see Section 4.5 and Appendix A). Sperm samples from *known* (*n* = 8) and *no-known* (*n* = 6) *paternity* donors were initially incubated in human tubal fluid (HTF) medium under either NC or CAP conditions, and each one was divided into two aliquots. The first aliquot was used to record BCECF basal fluorescence for 120 s, after which NH_4_Cl (20 mM) was added to artificially produce intracellular alkalinization (Figure 5a). To obtain absolute pH_i_ values, we placed the second aliquot in a calibration medium called H^+^Cal (pH 6.0), which contains the K^+^/H^+^ ionophore nigericin (10 µM) and a relatively high concentration of K^+^ (120 mM). Incubation of sperm cells in H^+^Cal causes the collapse of the H^+^ gradient across their PM, equalizing the pH_i_ with the extracellular pH (pH_e_) of the medium [47]. We then started recording BCECF fluorescence; after 120 s, we began sequential additions of KOH to elicit stepwise pH_e_ increases, which resulted in equivalent increases in pH_i_. These KOH additions were continued until a pH_e_ (and pH_i_) of 8.0 was reached (Figure 5b). To convert fluorescence data into absolute pH_i_ values, we followed the same protocol applied to *E*_m_ measurements (Section 2.1, Figure 2). Subpopulations of KOH-responding cells were selected (Figure 5b), and the median values obtained when BCECF fluorescence reached a plateau after each KOH addition were plotted against pH_e_ (Figure 5c). A linear model was adjusted to these data and used to obtain pH_i_ values by interpolation. Figure 5d shows the calculated pH_i_ inside the histograms comparing BCECF fluorescence values from each cell subpopulation indicated in Figure 5a,b. In the representative example shown, the pH_i_ of cells incubated under CAP conditions and after the NH_4_Cl stimulus were 6.76 and 7.32, respectively.

Following the same protocol detailed for *E*_m_ measurements, we evaluated the changes in pH_i_ upon capacitation in sperm samples from *known* (*n* = 8) and *no-known* (*n* = 6) *paternity* donors (Figure 5d). As expected, cytosolic alkalinization was observed in samples from *known paternity* donors upon capacitation, with an average difference of 0.36 pH units (6.58 ± 0.14 for NC versus 6.94 ± 0.12 for CAP conditions, *p* = 0.0078). Interestingly, we did not detect such a response when the average pH values for the *no-known paternity* samples were compared (6.67 ± 0.14 for NC versus 6.61 ± 0.08 for CAP conditions, *p* = 0.5469). To further analyze these data, we calculated the capacitation-induced pH_i_ increase (ΔpH_i_ = pH_iCAP_ − pH_iNC_) for each individual sample (Figure 5e). Similar to previous studies that have reported a pH_i_ increase upon sperm capacitation [19,20,21], we established a ΔpH_i_ = ±0.1 pH units as the threshold value to consider a sample as being alkalinized or acidified. On the basis of these criteria, we found that 100% (8/8) of the samples from *known paternity* donors exhibited the expected capacitation-associated pH_i_ alkalinization. In contrast, only one out of six (17%) of the samples from *no-known paternity* donors underwent pH_i_ alkalinization upon capacitation, while 33% of them remained unchanged and 50% became acidified (Figure 5f).

## 3. Discussion

The routine protocol used to assess sperm dysfunctions, the seminogram, does not encompass an evaluation of any of the capacitation-associated parameters required for fertilization (such as *E*_m_, pH_i_, and [Ca^2+^]_i_), even though their suitability as potential predictors of success during the implementation of ARTs has been recognized [25,26,27,28,29]. The experimental methods employed to assess these parameters include single-cell and population-based protocols, which present some disadvantages, including the inability to detect heterogeneous responses, technical challenges, and/or restrictions in the number of analyzed cells. To overcome such limitations, we propose the use of a retrofitted BD Accuri C6 Plus flow cytometer as a tool to evaluate physiological parameters related to human sperm capacitation. Our strategy has several advantages over existing methods: it is less time consuming; it allows the exclusion of non-viable cells; it evaluates a number of cells which far exceeds that of single-cell techniques; it records heterogeneous responses; and most importantly, it enables time-lapse fluorescence recordings, which makes possible the continuous detection of responses upon addition of different chemical stimuli. This is in sharp contrast with stopped-flow cytometry methods where data acquisition must be interrupted for test compound additions, leading to information losses, especially in the case of fast and transient responses [25].

Vines and colleagues first employed the BD Accuri C6 Plus to perform kinetic fluorescence recordings of rapid [Ca^2+^]_i_ changes in glioma cells [33]. Recently, Franchi and colleges applied for the first time this same approach to evaluate [Ca^2+^]_i_ changes in bovine sperm cells [34]. Here, we improved the implementation of the method (TLFC) by custom-building special glass tubes with a side opening to facilitate the continuous addition of test compounds. We also designed a 3D-printed special holder that enables placement of the sample tube on a magnetic stirrer to ensure a continuous homogenization of the cell suspension throughout the entire recording period.

The analysis of sperm *E*_m_ has gained clinical importance given that a high correlation between the sperm’s fertilizing potential and its ability to undergo capacitation-related PM hyperpolarization has been reported [17,25,26,27]. Here, after successful calibration of fluorescence data from Disc, we were able to obtain absolute *E*_m_ values for the first time using TLFC. We observed a wide range of variability in resting *E*_m_ values across samples, consistent with previously reported values for human sperm [25,26,27]. Furthermore, in this work we analyzed the capacitation-induced changes in resting *E*_m_ in cell samples from normozoospermic donors of *known* and *no-known paternity*. We observed that 100% of the samples from *known paternity* donors displayed the expected capacitation-associated *E*_m_ hyperpolarization, in contrast with only 44% of samples from *no-known paternity* donors. Despite this, when the average magnitude of the capacitation-induced change in resting E*_m_* was considered (−10 mV for samples from *known paternity* donors), our result was only slightly different from the −5 mV change that was reported when measured through patch-clamping (which is one of the most accurate methods to determine single-cell *E*_m_) [25]. It is worth mentioning that the average absolute resting *E*_m_ values thus far reported for human sperm under NC and CAP conditions varied considerably. For example, reported *E*_m_ resting values for NC and CAP conditions, respectively, were −10 to −40 mV (patch clamp [25]), −20 to −80 mV (stopped flow cytometry [27]), −44 to −74 mV [48], and −32 to −54.6 mV (spectrofluorometry [26]). The average *E*_m_ resting values obtained here by TLFC (considering samples from both *known* and *no-known paternity* donors*)* were −63 mV (NC) and −71 mV (CAP) (data not shown), which fell within the range of those measured through population-based approaches. Additionally, the difference we observed in resting *E*_m_ values between NC and CAP conditions (−11 mV) was similar to that reported from those methods (−9 mV for stopped-flow cytometry, and −10 mV for spectrofluorometry) [26,27]. Altogether, the data presented here support the use of TLFC as a robust tool to evaluate this very important parameter.

Considering that [Ca^2+^]_i_ signaling is fundamental for sperm capacitation, we evaluated the Pg-evoked Ca^2+^ influx using TLFC. In all samples, we were able to observe the expected increase in [Ca^2+^]_i_ upon Pg stimulation, consisting of a fast and transient [Ca^2+^]_i_ increase followed by a sustained plateau [24,43,44]. Stopped-flow cytometry has been used to evaluate this response with no success, since the fast [Ca^2+^]_i_ increase is lost due to recording interruption [49]. Both fluorescence microscopy and spectrofluorometry are the most common methods currently employed to evaluate [Ca^2+^]_i_ changes in sperm; as mentioned earlier, some of the disadvantages they present (a limitation in the number of cells that can be analyzed, and the impossibility of removing dead cells from the analysis, respectively) can be overcome by the use of TLFC. In this work, we decided to compare the [Ca^2+^]_i_ response upon Pg stimulation in sperm cells incubated either under NC or CAP conditions, since previous studies have shown that the [Ca^2+^]_i_ response increases by at least onefold upon capacitation [50,51,52,53,54]. As indicated before, all sperm samples displayed the typical Pg response; however, we observed high variability when comparing the magnitude of the [Ca^2+^]_i_ increase in NC versus CAP conditions. An increased response to Pg under CAP conditions was only observed for 61% of all samples analyzed (6 out of 7 from *known paternity* donors, and 5 out of 11 from *no-known paternity* donors). Previous reports suggest that the differences in responses after a Pg stimulus depend on the basal [Ca^2+^]_i_ of the sperm cells [29,55], which we did not analyze. Positive outcomes during ART procedures have been associated with a high increase in [Ca^2+^]_i_ after a Pg stimulus [29,30,56], while small or null Pg responses have been found in patients with unsuccessful outcomes from ART treatments [27] and in oligozoospermic individuals [57]. Moreover, the sperm’s egg-penetrating capability is also related to a high Pg response [55]. All these shreds of evidence indicate that the Pg-induced [Ca^2+^]_i_ responses are relevant to provide predictive values of fertility outcomes. Our results indicate that TLFC is completely suitable to analyze these responses, providing key advantages over other commonly employed techniques.

Another benefit of TLFC is the possibility of combining different dyes for the simultaneous evaluation of more than one physiological parameter. We co-stained sperm cells with Fluo3 and Disc in order to monitor [Ca^2+^]_i_ changes upon *E*_m_ manipulation. After valinomycin-evoked *E*_m_ hyperpolarization, a transient decrease in [Ca^2+^]_i_ was observed in most cases, and this was followed by a recovery phase that remained either below or similar to resting conditions. Then, a KCl-induced depolarization evoked a Ca^2+^ influx. These results are consistent with previous reports of voltage-dependent changes in [Ca^2+^]_i_ [52,58,59] that are presumably controlled by the CatSper channel, which presents a weak voltage dependency [23,24,60]. As mentioned before, this co-staining advantage of TLFC opens up the possibility of combining additional fluorescent dyes to further explore other physiological parameters in sperm. For instance, the simultaneous assessment of [Ca^2+^]_i_ changes and the sperm’s Pg-induced acrosome exocytosis could be performed using the appropriate combination of fluorescent labels (e.g., Fluo3 and a fluorescent-labeled lectin, respectively) [61]. This evaluation is of special interest, since the level of Pg-triggered acrosome exocytosis is yet another parameter associated with capacitation and fertility [30,62].

The rise in pH_i_ has been recognized as a hallmark of mammalian sperm capacitation [19,20,21]. Mice and human sperm cells possess proteins whose dysfunction produces alterations in pH_i_, and to some extent, impairs capacitation and reduces reproductive outcomes (reviewed in [45,46]). Here, we report the employment of TLFC as a robust tool to measure absolute pH_i_ values in human sperm cells. The range of the calculated resting pH_i_ values for the human sperm samples observed here (6.4 to 7.0) was within the reported range of pH_i_ values that have been obtained using different methods, including spectrofluorometry (6.9 to 7.1) [20], fluorescence microscopy (6.7) [47], and even stopped-flow cytometry (6.7 to 7.0) [28,63]. Recently, the capacitation-associated alkalinization has been investigated from a clinical standpoint, with results suggesting that sperm samples from patients with fertility issues present an impaired regulation of pH_i_ and fail to undergo the capacitation-associated alkalinization [28]. We thus explored the capacitation-associated alkalinization, which occurred consistently in all sperm samples from *known paternity* donors, but only in 17% of those from *no-known paternity* donors. A previous study reported that the average increase in pH_i_ after 24 h of capacitation was around 0.2 units for normozoospermic samples [20]. The average pH change we obtained for samples from *known paternity* donors was 0.4 units. Such difference in the capacitation-associated ΔpH_i_ could be explained partly by the difference in capacitation time (24 h vs. 6 h in our experiments), and by the natural variability that exists among individuals. More research is required to obtain further information about the fertilization potential of samples that do not display the capacitation-induced pH_i_ change. In summary, pH_i_ is another capacitation-related parameter with clinical relevance that can be readily analyzed by TLFC.

The results provided in this work support the notion that *E*_m_ hyperpolarization, [Ca^2+^]_i_ responses to Pg stimulation, and cytoplasmic alkalinization are physiological parameters that may convey information of the fertilizing potential of sperm samples. Further investigation is required to establish putative normal response ranges for these parameters, as well as to understand the clinical significance whenever a given sample fails to display a response within such ranges. Even though the BD Accuri C6 cytometer is not an instrument commonly found in fertility clinics at this time, we propose that it is a suitable and cost-effective tool to perform sperm TLFC recordings aimed at guiding ART selection. The instrument retrofitting required for this application is reversible and straightforward, involving the use of readily available and inexpensive materials. One of the disadvantages of using this equipment, however, is that the filter/illumination configuration is fixed and thus the dye choices are limited. Nevertheless, the evaluation of the three capacitation-associated parameters described here, and potentially additional ones, could be useful as additional pieces of evidence when assessing the fertilizing potential of a given sperm sample. Moreover, since our method evaluates several parameters related to molecular processes involved in sperm function, it offers the potential of providing new and valuable information about the specific mechanisms that may be altered in sperm samples from infertile men. As such, we believe that our methodology may open up the door to a novel and more specific way of diagnosing and treating male infertility.

## 4. Materials and Methods

### 4.1. Reagents

Chemicals were obtained from the indicated sources: 2′,7′-bis-(2-carboxyethyl)-5-(and-6)-carboxyfluorescein acetoxymethyl ester (BCECF-AM), 3,3′dipropylthiadicarbocyanine iodide (DiSC_3_(5)), and 4-(6-acetoxymethoxy-2,7-dichloro-3-oxo-9-xanthenyl)-4′-methyl-2,2′(ethylenedioxy) dianiline-N,N,N′,N′-tetra acetic acid tetrakis (acetoxymethyl) ester (Fluo3-AM) from ThermoFisher Scientific; ionomycin from Alomone; and all others from Sigma-Aldrich. Fluorescent dye and test compound stock solutions were prepared in Dimethyl sulfoxide (DMSO), except for those of KCl, KOH, NH_4_Cl, and MnCl_2_, which were prepared in distilled water.

### 4.2. Human Sperm Samples and Ethical Approval

Human sperm sample collection was approved by the Bioethics Committee of the Instituto de Biotecnología (UNAM, Cuernavaca, Morelos, México. Approval 368 from 1 January 2019 to 31 December 2021). Written informed consent was obtained from all sperm donors. Only those samples that fulfilled the WHO guidelines for normal semen parameters (i.e., normozoospermic) were used in this study (lower reference limit: semen volume = 1.5 mL, sperm number = 39 × 10^6^, normal morphology = 4%, total motility = 40%, pH = 7.2) [5]. Sperm samples were obtained from 2 types of normozoospermic donors: (i) *known paternity*, subjects with self-reported paternity, and (ii) *no-known paternity*, subjects without self-reported paternity and with no known fertility problems.

### 4.3. Human Sperm Sample Preparation

Human ejaculated sperm samples were collected by masturbation after 3 to 5 days of sexual abstinence. Semen samples were liquefied for 30 min at 37 °C in an atmosphere of 5% CO_2_; from this point onwards, all sample incubations were performed under these temperature and atmosphere conditions. Motile cells were recovered through the swim-up method (sperm cells are allowed to swim from the semen sample into a buffer aliquot layered above it; see [44] for details) either in non-capacitating (NC) 4-(2-hydroxyethyl)-1-piperazineethanesulfonic acid (HEPES)-buffered HTF medium (in mM: 90 NaCl, 4.68 KCl, 2.78 glucose, 1.8 CaCl_2_, 0.37 KH_2_PO_4_, 0.2 MgSO_4_, 0.33 sodium pyruvate, 21.39 sodium lactate, and 23.8 HEPES) or in capacitation-inducing (CAP) conditions, which consisted of HTF medium supplemented with 25 mM NaHCO_3_ and 0.5% (*w*/*v*) bovine serum albumin (BSA). Both media were adjusted to pH 7.4 with NaOH. After swim-up, sperm samples were incubated under NC or CAP conditions for 6 h. Cells under NC or CAP conditions were loaded with 2 µM Fluo3 for 30 min, or with 300 nM BCECF for 15 min protected from light. Excess dye was removed by centrifugation at 300× *g* for 5 min, and the pellet was resuspended in NC or CAP medium to obtain a sperm concentration of 3 × 10^6^ cells/mL. Finally, sperm cells were stained with 25 nM Disc for 10 min protected from light.

### 4.4. Experimental Setup to Perform TLFC

Vines et al. established the use of the BD Accuri C6 Plus flow cytometer (Figure 1a) to perform continuous monitoring of [Ca^2+^]_i_ in cells in suspension after sequential additions of test compounds [29]. We improved this strategy by using a custom-built system that facilitated the continuous addition of test compounds while ensuring their homogeneous mixing with the cell suspension and preventing their precipitation (Figure 1b), as many of them are dissolved in DMSO. We employed a modified glass tube with a side opening to ease the process of test compound solution additions by manual pipetting (Figure 1(b2,3)). We also designed and 3D-printed a special holder for these tubes (Figure 1(b5)). The modified glass tube containing the sperm cell suspension was placed on a magnetic stir plate (Four E’s Scientific, Guangzhou, China) for the continuous mixing (105 rpm) of the suspension during the entire data acquisition period (Figure 1(b4,6)). This entire setup is shown in Figure 1.

### 4.5. Data Acquisition and Selection of Viable Single Sperm Cells in the BD Accuri C6 Plus Flow Cytometer

We performed acquisitions of cellular events in a continuous mode to obtain time-lapse measurements with a flow rate of 14 µL/min. All acquisitions were performed at a room temperature of 20 °C. Fluorescence data were recorded as individual cellular events on a BD Accuri C6 Plus flow cytometer (Becton Dickinson, Franklin Lakes, NJ, USA). Forward scatter (FSC) and side-scatter (SSC) fluorescence data were collected for each sample. Threshold levels for FSC and SSC were established to exclude cellular debris (Appendix A), and a two-dimensional density plot of FSC height (FSC-H) versus FSC area (FSC-A) was used to eliminate cell aggregates from the analysis (Appendix A). Only single cells were included in further analyses. The cytometer acquisition settings were as follows: to detect BCECF and Fluo3, we employed the 488 nm laser as the excitation source; BCECF and Fluo3 emission signals were detected in the FL-1 channel set with a 533/30 filter; Disc fluorescence was excited with a 640 nm laser, and its emission was detected in the FL4-channel set with a 670 LP filter.

To select viable cells, we followed the strategy reported by Puga-Molina et al. [64]. Either Fluo3 or BCECF staining was employed as a viability marker, since these dyes are incorporated only by healthy cells, and are excluded from non-viable cells. This is illustrated for the case of Fluo3 in Appendix A. Heat-killed cells exhibit higher Fluo3 fluorescence intensity than unstained cells, but lower than viable cells. Therefore, we were able to establish a fluorescence intensity threshold to select viable cells for further analyses.

### 4.6. Measurement of Absolute E_m_ Values in Human Sperm

The fluorescence signal of sperm cells loaded with Fluo3 and Disc was recorded after Disc steady-state fluorescence was reached. The *E*_m_ fluorescence signal was calibrated by adding 1 µM valinomycin to induce PM hyperpolarization. Then, sequential additions of KCl were applied every 120 s to obtain final extracellular K^+^ concentrations of 5, 10, 15, 25, and 40 mM [16], corresponding to theoretical *E*_m_ values of −80, −63 −52 −40, and −28 mV, respectively. These theoretical *E*_m_ values were obtained using the Nernst equation, assuming an intracellular K^+^ concentration of 120 mM. To obtain the fluorescence values used to calculate the *E*_m_ values, we calculated the median fluorescence value of the responding sperm subpopulation when a plateau in the response was reached. The final sperm *E*_m_ value was then obtained by linear interpolation of the theoretical *E*_m_ values versus the median fluorescence values of each trace (see Section 2.1).

### 4.7. Evaluation of [Ca^2+^]_i_ Changes in Human Sperm

[Ca^2+^]_i_ was monitored through fluorescent measurements of the cell-permeable calcium indicator Fluo3. Upon cell entry, the acetoxymethyl ester group (AM) of the dye is hydrolyzed by cytosolic esterases, and free Fluo3 accumulates in the cytosol. The fluorescence intensity of Fluo3 exhibits an increase upon binding Ca^2+^. Loaded Fluo3 sperm cells under either NC or CAP conditions were registered for 600 s to evaluate [Ca^2+^]_i_ changes. At 120 s after fluorescence recording began, 4 µM Pg was added. To obtain the maximum fluorescence intensity, we added 10 µM ionomycin after recording for 300 s, followed by addition of 5 mM MnCl_2_ at 540 s to quench Fluo3 fluorescence and thus obtain a minimum fluorescence value.

### 4.8. Measurement of Absolute pH_i_ Values in Human Sperm

To determine pH_i_ values, we loaded sperm cells with BCECF, a pH-sensitive cell-permeable probe. Free BCECF accumulates in the cytosol in the same way that Fluo3. In the case of samples incubated in CAP medium, cells were resuspended in medium without BSA prior to dye loading. Once sperm cells (either in NC or CAP medium) were loaded with BCECF, the cell suspension was divided into 2 equal parts and the excess dye was removed by centrifugation for 5 min at 300 × *g*. One half of the suspension was incubated at 37 °C during at least 15 min in a calibration medium, herein named H^+^Cal (in mM: 120 KCl, 25 HEPES, 1 MgCl_2_, and 0.01 nigericin, adjusted to pH 6.0 with KOH). H^+^Cal medium allows the equilibration of pH_i_ with the pH_e_ through the effect of nigericin (an ionophore that facilitates K^+^/H^+^ exchange across the PM). The other half of the cells were resuspended in HTF medium under either NC or CAP conditions.

To measure pH_i_, we performed in vivo pH_i_ calibration. BCECF fluorescence of sperm cells incubated in the H^+^Cal (pH 6.0) medium was recorded for 120 s. Then, sequential additions of solutions with different KOH concentrations were performed every 120 s to raise the pH_e_ 0.25 units, until reaching a final pH_e_ of 8.0. The amount of KOH needed to raise the pH 0.25 units was calculated using the Henderson-Hasselbalch equation, considering a pKa = 7.55 for HEPES at 20 °C. The pH_i_ value of the sperm cell was obtained by recording the BCECF fluorescence of the cells resuspended in HTF medium under either NC or CAP conditions, and the absolute pH_i_ value for each sperm sample was calculated by fitting a logarithmic model of the median fluorescence values from the responding sperm subpopulation versus the pH_e_ value reached with each KOH addition. An NH_4_Cl was performed in each pH_i_ determination as a control of the responsiveness of the cells.

### 4.9. Statistical Analyses

Flow cytometry experimental data from the BD Accuri C6 Plus software were exported in Flow Cytometry Standard (FCS) format and initially analyzed with FlowJo Software version 10.1 (BD). Then, fluorescence and time data were exported into a comma-separated values (CSV) format and analyzed in RStudio version 1.3.1093 [65]. Fluorescence data are shown as density plots where warm colors represent high cell density. Black lines in density plots represent smoothed traces of each recording, which corresponds to the median fluorescence value for each given time point (obtained in FlowJo with the Kinetics function). Graphs were created with ggplot2 and ggpubr [66]. The results obtained for *E*_m_ values, Pg-triggered peak [Ca^2+^]_i_ fluorescence, and pH_i_ values were first separated into the 2 donor classes—*known paternity* and *no-known paternity*. In each of these groups, paired comparisons were performed between NC and CAP conditions by a paired *t*-test or a Wilcoxson signed-rank test, depending on whether or not the data passed the Shapiro-Wilk normality test, respectively. After calculating the Δ*E*_m_ (*E*_mCAP_ − *E*_mNC_), ΔF_Peak_ (F_PeakCAP_ − F_PeakNC_), and ΔpH_i_ (pH_iCAP_ − pH_iNC_), we performed a one-sample *t*-test comparison against theoretical mean = 0 for each donor class—*known paternity* and *no-known paternity* (calculations were performed in GraphPad Prism 8). A probability value (*p*) < 0.05 was considered as a statistically significant difference. The final versions of the figures were made in Inkscape 0.91 (Inkscape.org).

## Figures and Tables

**Figure 1 ijms-22-00093-f001:**
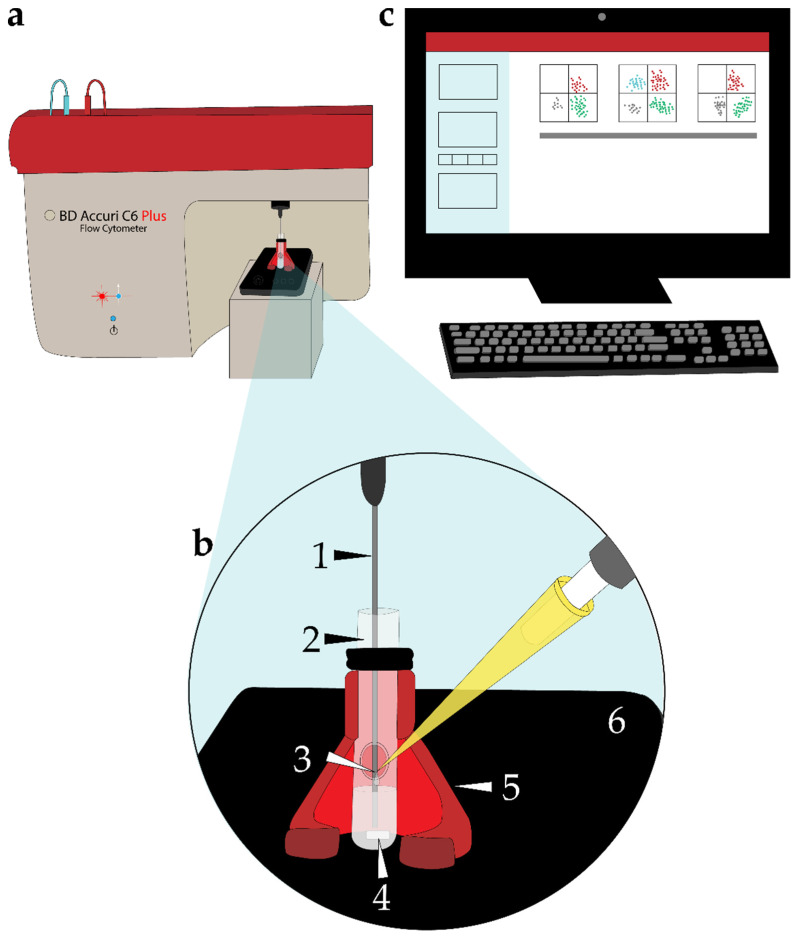
Schematic diagram of the time-lapse flow cytometry (TLFC) instrument setup. (**a**) BD Accuri C6 Plus flow cytometer retrofitted with a custom-built time-lapse measurement system. (**b**) Detail of the custom-built component to perform continuous reagent additions and to ensure homogeneous sample mixing during data acquisition. (1) Sample injection port (SIP) consisting of a sample injection tube through which the sample travels to the flow cell to be introduced to the sheath fluid. (2) Modified glass tube with a port (side opening) to enable the continuous addition of reagents to the cell suspension. (3) Reagent addition to the cell suspension was accomplished through micropipetting. (4) Magnetic stir bar. (5) 3D-printed tube holder specifically built to accommodate the modified glass tube. (6) Magnetic stir plate for the continuous mixing of the cell suspension throughout the entire data acquisition period. (**c**) Desktop computer running the BD Accuri C6 Plus software program.

**Figure 2 ijms-22-00093-f002:**
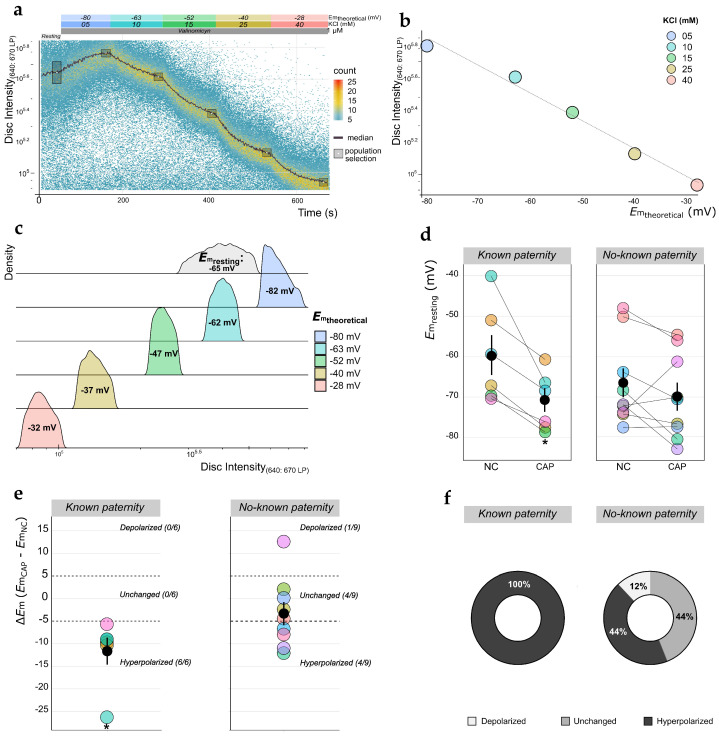
Measurement of membrane potential (*E*_m_) absolute values in human sperm. (**a**) Representative time-lapse flow cytometry (TLFC) 3,3′dipropylthiadicarbocyanine iodide (DiSC_3_(5), abbreviated to Disc) fluorescence density plot of a sperm sample from a *known paternity* donor under capacitating (CAP) conditions. Colored boxes above the plot indicate the nature and duration of the employed stimuli. The dark solid line indicates the median fluorescence value. Gray boxes enclose the cell subpopulations selected for further calculations. (**b**) Median fluorescence values for the selected cell subpopulations indicated in (**a**) were plotted against the theoretical *E*_m_ value (calculated using Nernst’s equation). The dashed line indicates the linear fit (*y* = −10,682.4x − 264,308; *R*^2^ = 0.95; *p* = 0.003). (**c**) Density histograms of Disc fluorescence values from the subpopulations indicated in (**a**). For comparison, each colored histogram corresponds to each of the *E*_m_ values (indicated inside histograms) estimated using the linear fit in (**a**); the interpolated resting *E*_m_ value (gray histogram) of the resting sperm subpopulation was −65 mV. (**d**) Plots of calculated *E*_m_ values for individual sperm samples (shown color-coded) incubated under either non-capacitating (NC) or CAP conditions, from either *known* (*n* = 6) or *no-known* (*n* = 9) *paternity* donors. Black dots and solid lines indicate mean ± standard error of the mean (SEM), respectively; * *p* < 0.05 against NC according to a paired *t*-test. (**e**) Plots of capacitation-induced *E*_m_ change (Δ*E*_m_ = *E*_mCAP_ − *E*_mNC_) for samples in (**d**). Samples with a Δ*E*_m_ value greater, lower, or equal to ±5 mV were classified (dotted lines) as depolarized, hyperpolarized, or unchanged, respectively; numbers in parentheses indicate the number of samples in each category out of the total. Black dots and solid lines indicate mean ± SEM; * *p* < 0.05 according to a one-sample *t*-test comparison against a theoretical mean of Δ*E*_m_ = 0. (**f**) Donut charts depicting the distribution of the results shown in (**e**).

**Figure 3 ijms-22-00093-f003:**
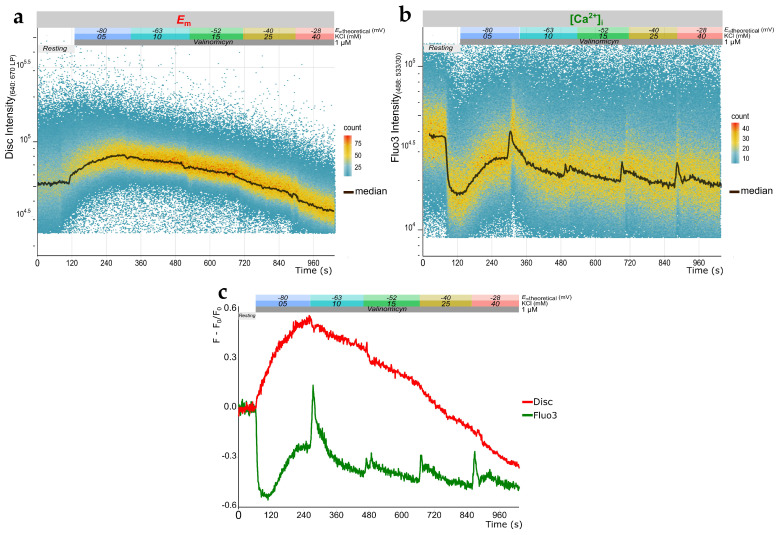
Multi-parametric measurement of membrane potential (*E*_m_) and intracellular calcium concentration ([Ca^2+^]_i_) changes in human sperm. Representative time-lapse flow cytometry (TLFC) 4-(6-acetoxymethoxy-2,7-dichloro-3-oxo-9-xanthenyl)-4′-methyl-2,2′(ethylenedioxy) dianiline-N,N,N′,N′-tetra acetic acid tetrakis (acetoxymethyl) ester (Fluo3-AM, abbreviated to Fluo3) fluorescence density plots of a sperm sample from a *known paternity* donor, which was co-stained with DiSC_3_(5) (Disc) (**a**) and Fluo3 (**b**) to analyze *E*_m_ and [Ca^2+^]_i_ changes, respectively, under capacitating conditions. Colored boxes above the plots indicate the nature and duration of the employed stimuli. (**c**) Normalized median fluorescence values ((F − F_0_)/F_0_) of data shown in (**a**) and (**b**).

**Figure 4 ijms-22-00093-f004:**
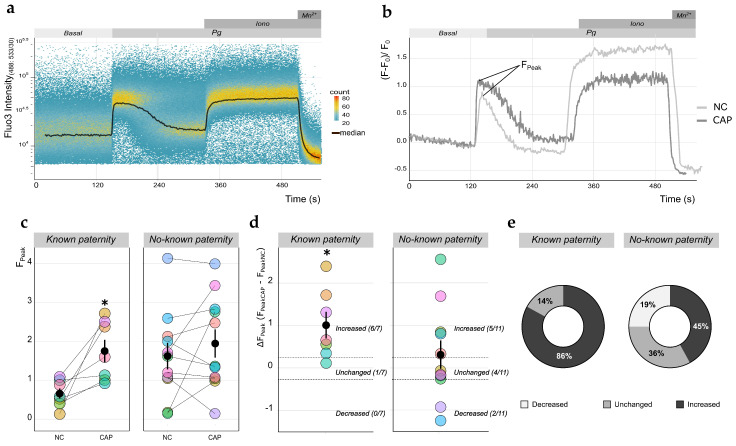
Qualitative evaluation of progesterone (Pg)-induced intracellular calcium concentration [Ca^2+^]_i_ changes in human sperm. (**a**) Representative time-lapse flow cytometry (TLFC) Fluo3-AM (Fluo3) fluorescence density plot of a sperm sample from a *known paternity* donor under capacitation conditions. Gray boxes above the plot indicate the nature and duration of the employed stimuli: progesterone (Pg), ionomycin (Iono), and manganese chloride (Mn^2+^). The dark solid line indicates the median fluorescence value. (**b**) Representative example of normalized median fluorescence values ((F − F_0_)/F_0_) for an entire sperm population incubated under non-capacitating (NC) or capacitating (CAP) conditions. The F_Peak_ is indicated for both traces. Gray boxes above the plot indicate the nature and duration of the employed stimuli. (**c**) Comparison of F_Peak_ responses for individual sperm samples (shown color-coded) from *known* (*n* = 7) and *no-known* (*n* = 11) *paternity* donors, incubated either under NC or CAP conditions. Black dots and solid lines indicate mean ± SEM; * *p* < 0.05 according to a paired *t*-test. (**d**) Plots of the difference in F_Peak_ between NC and CAP conditions (ΔF_Peak_ = F_PeakCAP_ − F_PeakNC_) for samples in (**c**). Samples with a ΔF_Peak_ greater, lower, or equal to ±0.25 were classified (dotted lines) as increased, decreased, or unchanged response, respectively; numbers in parentheses indicate the number of samples in each category out of the total. Black dots and solid lines indicate mean ± SEM; * *p* < 0.05 according to a one-sample *t*-test comparison against a theoretical mean of ΔF_Peak_ = 0. (**e**) Donut charts depicting the distribution of the results shown in (**d**).

**Figure 5 ijms-22-00093-f005:**
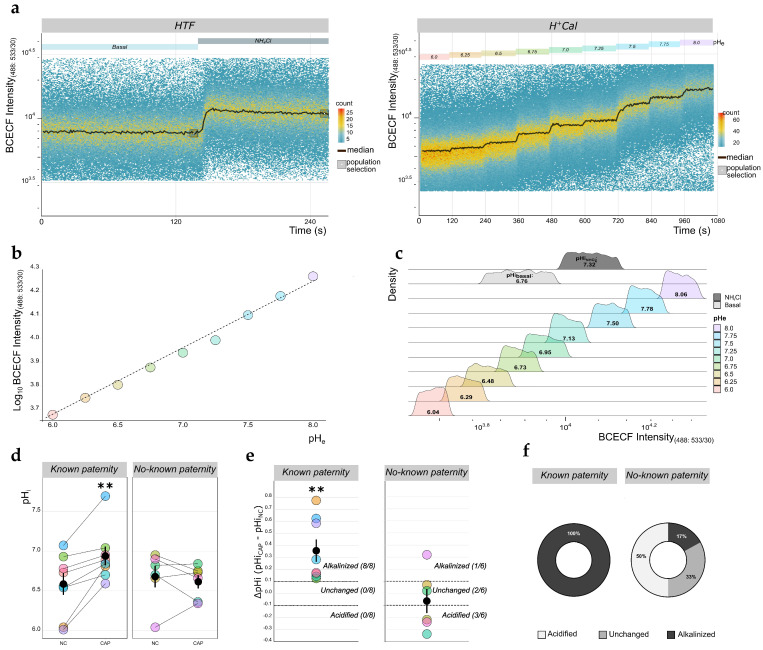
Measurement of intracellular pH (pH_i_) absolute values in human sperm. (**a**) Representative time-lapse flow cytometry (TLFC) 2′,7′-bis-(2-carboxyethyl)-5-(and-6)-carboxyfluorescein acetoxymethyl ester (BCECF-AM, abbreviated to BCECF) fluorescence density plots of a sperm sample from a *known paternity* donor incubated under non-capacitating (NC) conditions in either human tubal fluid (HTF) (left panel) or calibration (H^+^Cal) (right panel) medium. Colored boxes above the plots indicate the nature and duration of the employed stimuli. The dark solid lines indicate the median fluorescence values. Gray boxes enclose the cell subpopulations selected to perform further calculations. (**b**) Median log_10_-transformed BCECF fluorescence values of the selected subpopulations indicated in (**a**) (right panel) plotted against pH_e_. The dashed line indicates the linear fit model (*y* = −0.29x − 1.93; *R*^2^ = 0.99; *p* = 7.5 × 10^−9^). (**c**). Density histograms of BCECF fluorescence values from the selected cell subpopulations indicated in (**a**) (right panel). For comparison, each colored histogram corresponds to each of the estimated pH_i_ values (using the Henderson-Hasselbalch equation); the interpolated basal (light gray) and NH_4_Cl-alkalinized (dark gray) pH values (indicated inside histograms) were 6.76 and 7.32, respectively. (**d**) Plots of estimated pH_i_ values for individual sperm samples (shown color-coded) incubated under either NC or capacitating (CAP) conditions, from either *known* (*n* = 8) or *no-known* (*n* = 6) *paternity* donors. Black dots and solid lines indicate mean ± SEM, respectively; ** *p* < 0.01 against NC conditions according to a Wilcoxon matched-pairs signed-rank test. (**e**) Plots of the capacitation-induced pH_i_ change (ΔpH_i_ = pH_iCAP_ − pH_iNC_) for samples in (**d**). Samples with a ΔpH_i_ greater, lower, or equal to ±0.1 pH_i_ units were classified (dotted lines) as alkalinized, acidified, or unchanged, respectively; numbers in parentheses indicate the number of samples in each category out of the total. Black dots and solid lines indicate mean ± SEM, respectively; ** *p* < 0.01 according to a one-sample *t*-test comparison against a theoretical mean of ΔpH_i_ = 0. (**f**) Donut charts depicting the distribution of the results shown in (**e**).

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
