# Peer review of "Time-Lapse Flow Cytometry: A Robust Tool to Assess Physiological Parameters Related to the Fertilizing Capability of Human Sperm"

_ijms, 2020, doi:10.3390/ijms22010093_

Round 1

Reviewer 1 Report

The manuscript entitled Time-lapse flow cytometry: A robust tool..." by Matamoros-Volante et al. is very well written and the premise of this study is well explained to the readers of all kinds. The only issue that I was surprised by is the amount of methodology that is presented in the results. While some of the methods in the results could be removed to the Material and Methods section, I do not think it is critical enough given that the study itself is 'methodology development'. I would leave this issue to either the authors themselves or the editors.

The elegance of the methods presented for the assessment of capacitation related human spermatozoa pathology is in my opinion quite reasonable enough for clinical adoption. This makes the approval of this manuscript for publication a no-brainer. There are barely few notes that I have for the authors to consider or explain. Many of these could be taken care of in the manuscript proof or by the editors themselves. They are as below:

Lines 87-99 Readability - repeated emphasis on 'three capacitation parameters'. This paragraph could use some consolidation of sentences and ideas.

Throughout the manuscript the membrane potential Em must be uniformly in the format of Italicized E and normal but subscripted 'm'… Em

I am not sure about IJMS requirements, but I believe the widely used abbreviation for progesterone is either P or P4 and not Pg.

Figure 1: Labeling of the custom built component could be changed from Roman numerals to either numbers or letters or abbreviations such as MS: magnetic stir bar; SIP: Sample injection port... etc.

L.135 : Nest's equation must be Nernst's equation

Use of magnetic stir bar: Would the authors add the RPM for stir bar in the semen sample  - I believe it is not mentioned in the manuscript?

Related to the use of a stir bar, have the authors noted any change in the spermatozoa viability due to the use of stir bar pre- flow cytometry  ?

Author Response

The manuscript entitled Time-lapse flow cytometry: A robust tool..." by Matamoros-Volante et al. is very well written and the premise of this study is well explained to the readers of all kinds. The only issue that I was surprised by is the amount of methodology that is presented in the results. While some of the methods in the results could be removed to the Material and Methods section, I do not think it is critical enough given that the study itself is 'methodology development'. I would leave this issue to either the authors themselves or the editors.

R: We agree with the reviewer about the fact that we included a large amount of methodology in the Results section, and we appreciate their acknowledgment of this being a methodology development paper. Thus, we would prefer to leave the amount of information as is, provided the editors agree.

The elegance of the methods presented for the assessment of capacitation related human spermatozoa pathology is in my opinion quite reasonable enough for clinical adoption. This makes the approval of this manuscript for publication a no-brainer. There are barely few notes that I have for the authors to consider or explain. Many of these could be taken care of in the manuscript proof or by the editors themselves. They are as below:

Lines 87-99 Readability - repeated emphasis on 'three capacitation parameters'. This paragraph could use some consolidation of sentences and ideas.

R: We appreciate this suggestion, and we have edited the paragraph accordingly, eliminating unnecessary repetition.

Throughout the manuscript the membrane potential Em must be uniformly in the format of Italicized E and normal but subscripted 'm'… Em

R: We have made the necessary changes to homogenize the format of this abbreviation both in the main text and in the figures.

I am not sure about IJMS requirements, but I believe the widely used abbreviation for progesterone is either P or P4 and not Pg.

R: Although P and P4 are common abbreviations for progesterone, Pg is also widely used, so we have left it as is, but we would be happy to change it if required by IJMS.

Figure 1: Labeling of the custom built component could be changed from Roman numerals to either numbers or letters or abbreviations such as MS: magnetic stir bar; SIP: Sample injection port... etc.

R: We appreciate this suggestion, and we have changed them to arabic numerals (we were uncertain about using them since they are employed for figure numbering as well, but we prefer them as well, provided IJMS approves). Taking advantage of this editing opportunity, we have switched the order of labels #4 and #5, so that they are now easier to locate in the figure (counterclockwise).

L.135 : Nest's equation must be Nernst's equation

R: We have made the correction, thank you!

 Use of magnetic stir bar: Would the authors add the RPM for stir bar in the semen sample  - I believe it is not mentioned in the manuscript?

R: This was an oversight on our part, and we have added the rpm value (105 rpm) to Materials & Methods Section 4.4.

 Related to the use of a stir bar, have the authors noted any change in the spermatozoa viability due to the use of stir bar pre- flow cytometry 

R: This is an important concern, but the agitation is very gentle and we have not noticed any damage to the cells. Nonetheless, our method enables the exclusion of nonviable cells from the analysis; the details can be found in Figure S1.

Reviewer 2 Report

This Ms reports the use of TLFC for simultaneous evaluation of  functional changes related to sperm fertilization potential in human.

The Ms is novel and interesting.

I have some comments

Line 43  For not expert readership should be important to introduce the general concept of gametes and fertilization. In case of seminogram briefly quote what are the   main  cut off values as for example over 15 ML/ml for sperm concentration, over 4% for normal morphology etc..

Line 51 Similarly,  sperm capacitation is a very important issue sometime difficult to understand even for expert researchers . Authors must better introduce this concept by also quoting the review by Tosti and Menezo, HRU 2016.

Line 55 better describe the significance of plasma membrane potential and hyperpolarization. Authors can refer to the review by Tosti and Boni, HRU 2004. 

Line 56 Clarify that capacitation induced by Progesterone is one among the many stimuli.  In line 200 authors explain well its use and that of ionomicin. Although I agree with the authors that Pg is the physiological inducer of capacitation, explain also why they have not considered the use of calcium ionophore A23187 which is the historical compound used in the ARIC test to visualize the sperm capability to undergo in acrosome reaction.

Line 68 better to say the electrophysiological technique of patch clamp. Here the reference 23 is not fully appropriated, better to add one of the  numerous papers by Darszon.

Line 95-96 and 440-442. I do not understand why authors have not tested this method also on spermatozoa from  infertile men. Many of them approaching ARTs have good sperm parameters based on WHO guideline but are not able to get a pregnancy.  In this work  the comparison procedure would have been more explicative and complete.

Line 315 Do not start the discussion saying “Unfortunately”. Authors can define a limitation  in the use of the TLFC, since limits exist for most of the experimental setup.

Line 447 describe briefly the swim-up method

In conclusion I find the work very intriguing. The experimental part is complex and well designed. The Ms is well written and the results are convincing.

Among the valuable advantages of using TLFC  are included the co-staining, the exclusion of non viable sperm and the opportunity of test different stimuli.

This new tool, nonetheless  the limitations described at the beginning of discussion may be beneficial and relevant in the clinical practice  for screening an infertile population of spermatozoa that show normal physiological parameters at the first glance at the optical microscope.

I support the publication of this Ms after minor revisions

Author Response

This Ms reports the use of TLFC for simultaneous evaluation of  functional changes related to sperm fertilization potential in human.

The Ms is novel and interesting.

I have some comments

Line 43  For not expert readership should be important to introduce the general concept of gametes and fertilization. 

R: We appreciate this suggestion, and we have included the following sentence at the beginning of that first paragraph of the Introduction section, which now reads as follows: 

“Sexual reproduction involves the successful fusion of the female (egg or oocyte) and male (sperm) gametes, a process called fertilization. Human fertility issues are a growing public health problem worldwide [1]”

In case of seminogram briefly quote what are the   main  cut off values as for example over 15 ML/ml for sperm concentration, over 4% for normal morphology etc..

R: We have included in Materials & Methods Section 4.2 the parameters established by the WHO to consider a sample as normozoospermic, as follows:

“(lower reference limit: semen volume = 1.5 mL, sperm number = 39x106, normal morphology = 4%, total motility = 40%, pH = 7.2)”

Line 51 Similarly,  sperm capacitation is a very important issue sometime difficult to understand even for expert researchers . Authors must better introduce this concept by also quoting the review by Tosti and Menezo, HRU 2016.

R: We agree with this assessment, and we have included a more detailed capacitation definition using the suggested reference. 

Line 55 better describe the significance of plasma membrane potential and hyperpolarization. Authors can refer to the review by Tosti and Boni, HRU 2004. 

R: We have included a reference to the suggested review, so that readers may find more detailed information.

Line 56 Clarify that capacitation induced by Progesterone is one among the many stimuli.  In line 200 authors explain well its use and that of ionomicin. Although I agree with the authors that Pg is the physiological inducer of capacitation, explain also why they have not considered the use of calcium ionophore A23187 which is the historical compound used in the ARIC test to visualize the sperm capability to undergo in acrosome reaction.

R: The reviewer is right, A23187 is a calcium ionophore commonly used to induce the acrosome reaction, which we did not measure. We decided to use progesterone, both because it represents an actual physiological stimulus, and because at this time we were only interested in the [Ca2+]i increase it produces, which is linked to a greater success rate in ARTs. In the future, however, it would be informative to also measure the acrosome reaction using A2318.

Line 68 better to say the electrophysiological technique of patch clamp. Here the reference 23 is not fully appropriated, better to add one of the  numerous papers by Darszon.

R: We appreciate these suggestions and have changed the name of the technique, and added the following references: Darszon et al., 2005 and Kirichok et al., 2011. We did keep reference 23 (now 25), as it is appropriate in the sense that it refers to the relevance of membrane potential measured by patch clamp and its relation to ARTs success rate. 

Line 95-96 and 440-442. I do not understand why authors have not tested this method also on spermatozoa from  infertile men. Many of them approaching ARTs have good sperm parameters based on WHO guideline but are not able to get a pregnancy.  In this work  the comparison procedure would have been more explicative and complete.

R: Ideally, we would have indeed preferred to include sperm samples from infertile men. Unfortunately, we did not have access to such samples because local fertility clinics are currently closed due to the pandemic. We decided nonetheless to proceed with the limited samples available to us, given that the purpose of this initial paper is to describe the methodology in detail. The next step would be to validate it on sperm samples from patients seeking ART procedures. 

Line 315 Do not start the discussion saying “Unfortunately”. Authors can define a limitation  in the use of the TLFC, since limits exist for most of the experimental setup.

R: We appreciate these suggestions, and have eliminated the word “Unfortunately” from that sentence. We have also modified the final part of the last paragraph of the Discussion section to include some of the current limitations of our proposed method, which now reads as follows:

“Even though the BD Accuri C6 cytometer is not an instrument commonly found in fertility clinics at this time, we propose that it is a suitable and cost-effective tool to perform sperm TLFC recordings aimed at guiding ART selection. The instrument retrofitting required for this application is reversible and straightforward, involving the use of readily available and inexpensive materials. One of the disadvantages of using this equipment, however, is that the filter/illumination configuration is fixed and thus the dye choices are limited. Nevertheless, the evaluation of the three capacitation-associated parameters described here, and potentially additional ones, could be useful as additional pieces of evidence when assessing the fertilizing potential of a given sperm sample.”

Line 447 describe briefly the swim-up method

R: A brief description of the swim-up method and a reference is now included in Materials and Methods section 4.3, as follows:

“(sperm cells are allowed to swim from the semen sample into a buffer aliquot layered above it; see 44 for details)” 

In conclusion I find the work very intriguing. The experimental part is complex and well designed. The Ms is well written and the results are convincing.

Among the valuable advantages of using TLFC  are included the co-staining, the exclusion of non viable sperm and the opportunity of test different stimuli.

This new tool, nonetheless  the limitations described at the beginning of discussion may be beneficial and relevant in the clinical practice  for screening an infertile population of spermatozoa that show normal physiological parameters at the first glance at the optical microscope.

I support the publication of this Ms after minor revisions

R: We truly appreciate such positive and encouraging feedback.

Reviewer 3 Report

In this manuscript, the authors developed a novel strategy employing time-lapse flow cytometry (TLFC) to determine capacitation-related membrane potential (Em) and pHi changes, and progesterone-induced [Ca2+]i increases. Furthermore, they used spermatozoa from two types of normozoospermic donors: known paternity (subjects with self-reported paternity) and no-known 25 paternity (subjects without self-reported paternity and no known fertility problems) to support the usefulness of this new methodology.

The study design is interesting. The strength of this manuscript is the elucidate the advantage of this new method for detecting transient physiological response by flowcytometry. However, the lack of male factor infertility for the control group may decrease the clinical applicability of this new method. According to the design of the manuscript, the method may be applied to couples with unexplained infertility only. Under such circumstance, this TLFC needs a lot of investigation for use in male fertility evaluation and ART selection.

Furthermore, all the measured events, such as membrane potential (Em) and pHi changes, and progesterone-induced [Ca2+]i increases, are capacitation related responses. Induced acrosome reaction, hyperactivation by CASA, zona-binding assay, and hyaluronan binding assay (HBA) are clinically approved methods to evaluate the capacitation of spermatozoa. Does this new method provide new information in addition to those assays?

Author Response

In this manuscript, the authors developed a novel strategy employing time-lapse flow cytometry (TLFC) to determine capacitation-related membrane potential (Em) and pHi changes, and progesterone-induced [Ca2+]i increases. Furthermore, they used spermatozoa from two types of normozoospermic donors: known paternity (subjects with self-reported paternity) and no-known 25 paternity (subjects without self-reported paternity and no known fertility problems) to support the usefulness of this new methodology.

The study design is interesting. The strength of this manuscript is the elucidate the advantage of this new method for detecting transient physiological response by flowcytometry. However, the lack of male factor infertility for the control group may decrease the clinical applicability of this new method. According to the design of the manuscript, the method may be applied to couples with unexplained infertility only. Under such circumstance, this TLFC needs a lot of investigation for use in male fertility evaluation and ART selection.

R: We do recognize that the use of sperm samples from infertile men would have been ideal. Unfortunately, we did not have access to such samples because local fertility clinics are currently closed due to the pandemic. We decided nonetheless to proceed with the limited samples available to us, given that the purpose of this initial paper is to describe the methodology in detail. The next step would be to validate it on sperm samples from patients seeking ART procedures. 

Furthermore, all the measured events, such as membrane potential (Em) and pHi changes, and progesterone-induced [Ca2+]i increases, are capacitation related responses. Induced acrosome reaction, hyperactivation by CASA, zona-binding assay, and hyaluronan binding assay (HBA) are clinically approved methods to evaluate the capacitation of spermatozoa. Does this new method provide new information in addition to those assays?

R: This is a very interesting question to ponder, and it has prompted us to add some pertinent comments to our manuscript. Our method evaluates sperm parameters involving molecular mechanisms related to sperm function, as opposed to making end-point measurements such as hyperactivation, acrosome reaction, and HBA. By focusing on the underlying molecular processes, our methodology does provide new and valuable information about the specific mechanisms that may be altered in sperm samples from infertile men. We have added the following sentences at the end of the Discussion section:

“ Even though the BD Accuri C6 cytometer is not an instrument commonly found in fertility clinics at this time, we propose that it is a suitable and cost-effective tool to perform sperm TLFC recordings aimed at guiding ART selection. The instrument retrofitting required for this application is reversible and straightforward, involving the use of readily available and inexpensive materials. One of the disadvantages of using this equipment, however, is that the filter/illumination configuration is fixed and thus the dye choices are limited. Nevertheless, the evaluation of the three capacitation-associated parameters described here, and potentially additional ones, could be useful as additional pieces of evidence when assessing the fertilizing potential of a given sperm sample. Moreover, since our method evaluates several parameters related to molecular processes involved in sperm function, it offers the potential of providing new and valuable information about the specific mechanisms that may be altered in sperm samples from infertile men. As such, we believe that our methodology may open up the door to a novel and more specific way of diagnosing and treating male infertility.”